# Transactional Interpretation and the Generalized Poisson Distribution

**DOI:** 10.3390/e24101416

**Published:** 2022-10-04

**Authors:** Marcin Makowski, Edward Wiktor Piotrowski

**Affiliations:** Faculty of Physics, Department of Mathematical Methods in Physics, University of Białystok, Ul. Ciołkowskiego 1L, 15-245 Białystok, Poland

**Keywords:** squeezed coherent states, Fisher information, Fourier transform, Schrödinger-like equation, market, risk, supply and demand, quantum computer

## Abstract

The aim of this paper is to study the quantum-like approach to the description of the market in the context of the principle of minimum Fisher information. We wish to investigate the validity of using squeezed coherent states as market strategies. For this purpose, we focus on the representation of any squeezed coherent state with respect to the basis of the eigenvectors of the observable of market risk. We derive a formula for the probability of being the squeezed coherent state in one of these states. The distribution that we call generalized Poisson establishes the relation between the squeezed coherent states and their description in the language of risk in quantum terms. We provide a formula specifying the total risk of squeezed coherent strategy. Then, we propose a *risk of risk* concept that is in fact the second central moment of the generalized Poisson distribution. This is an important numerical characterization of squeezed coherent strategies. We provide its interpretations on the basis of the uncertainty relation for time and energy.

## 1. Introduction

Developments in quantum computing and quantum information theory have helped extend the scope of game theory to the quantum world [1,2,3,4]. This, in turn, has led to attempts at a quantum description of the market, due to the need to develop better methods of analyzing the dynamics of markets. It seems particularly important to create more accurate methods of forecasting extreme events such as crises or speculative bubbles. The history of recent crises shows that the methods developed in the framework of classical economics have failed. It is natural to believe that new methods can be provided by quantum mechanics [5], which has radically changed the way we perceive the world. In order to describe the quantum characters of the market, we are going to build a mathematical model on the basic postulates of quantum mechanics. The first of these postulates is that the state of a quantum mechanical system is completely specified by the wave function. By applying this postulate, the fluctuation of the stock price can be viewed as the evolution of the wave function. This evolution would be characterized by a dynamical equation, such as the Schrödinger equation. In this work, we refer to the quantum description of the market based on quantum game theory.

One direction of such analysis is the description of market transactions in terms of supply and demand curves [6,7,8,9,10]. In this approach, quantum strategies are vectors in a certain Hilbert space and can be interpreted as superpositions of trading decisions. The set of all such strategies defines all possible movements of market participants that can be implemented on a physical device such as a quantum computer.

More precisely, strategies of market participants are represented by wave functions. There are two basic representations: demand ψ(x) and supply ψ(y), where *x* and *y* are values of random variables representing logarithms of prices at which players buy or sell. Cumulative values of squares of modules of market strategies ${\left|\psi \left(x\right)\right|}^{2}$ and ${\left|\psi \left(y\right)\right|}^{2}$ correspond to probabilistic demand or supply curves. To describe the dynamics resulting from market participants’ tactics that modify their behaviour in the market, we use unitary operations on the Hilbert space—the strategy space of square-integrable functions $L^{2}$. The basic tactic for changing the representation of a strategy from supply to demand and vice versa is described by a Fourier transform (FT):ψ(y):=FTψ(x).
There are also tactics that only modify the demand or supply strategy without changing sides. The above formalism can be presented in an elegant way with the help of the Wigner functions defined on the common domain of variables *x* and *y* (the phase space):$$f_{n}\left(x,y\right)=\frac{1}{2\pi }\int_{-\infty }^{\infty }\psi_{n}\left(x+\frac{s}{2}\right)\;\psi_{n}\left(x-\frac{s}{2}\right)\;\cos\left(s\;y\right)ds\;.$$
Conditional (fixed public price for buying or selling) demand and supply curves are depicted by the graphs of the following *CDFs*:$$CDF_{d}\left(\ln\;c\right)=\int_{-\infty }^{\ln\;c}f_{n}\left(x=const.,y\right)dy,$$
$$CDF_{s}\left(\ln\;c\right)=\int_{-\infty }^{\ln\;\frac{1}{c}}f_{n}\left(x,y=const.\right)dx,$$
where *c* denotes the price of the good in question. This formalism is convenient for analyzing exceptions to the classical laws of supply and demand (Giffen goods), which we can view as negative probabilities [10].

An important parameter that plays a key role in making decisions on the market is the risk associated with a specific strategy. We define risk operators for both the demand representation and the supply representation
$$R_{d}\psi \left(x\right):=x^{2}\psi \left(x\right),\;\;\;\;\;\;\;\;R_{s}\psi \left(y\right):=y^{2}\psi \left(y\right)\;.$$
We call them this because of their average values
$$\langle R_{d}\rangle =\int_{-\infty }^{\infty }x^{2}\;\psi^{2}\left(x\right)\;dx,\;\;\;\langle R_{s}\rangle =\int_{-\infty }^{\infty }y^{2}\;\psi^{2}\left(y\right)\;dy$$
correspond to the variance of the random variables *x* and *y*. This is a popular measure of risk used in financial mathematics. Such a supply–demand perception of the market indicates its connection with the principle of minimum Fisher information [11,12]. We must remember that all games played in the real world must be implemented on the basis of physical processes. Therefore, the fundamental quantum constraints known to us should also be barriers respected by any complete game theory.

The article is organized as follows. In Section 2, we present the idea of a transactional interpretation for the Principle of Minimum Fisher Information, an important element of which is the so-called risk balance equation. This work was intended as a continuation of this idea, so we briefly sketch it. More details can be found in [13]. In the next Section, we discuss coherent states, which are essential for us to consider. It is not our purpose to study this topic in detail. This is only a brief introduction to it and may be of help to readers unfamiliar with the topic. We will restrict our attention to the definition of coherent states as states minimizing the Heisenberg uncertainty principle. In Section 4, we find a representation of a squeezed coherent state strategy on the basis of the eigenvectors of the observable of market risk. In Section 5, we derive an interesting formula for the probability of being the squeezed coherent state in one of the basis vectors. Section 6 contains a brief summary. In this section, we provide a formula specifying the total risk of the squeezed coherent strategy and we introduce a risk of risk concept. The interpretation of this numeric value is provided in the last section. Our viewpoint sheds some new light on the concept of risk measurement in quantum terms.

## 2. A Brief Introduction to Transactional Interpretation for the Principle of Minimum Fisher Information

The Fisher information $I_{F_{\theta }}$ is a measure of the amount of information carried by the observable random variable *X* about an unknown parameter θ of the probability distribution f(x;θ) modeling *X*. It is defined as
$$I_{F_{\theta }}=E\left[\left.{\left(\frac{\partial }{\partial \theta }\log f\left(X;\theta \right)\right)}^{2}\right|\theta \right]=\int_{-\infty }^{\infty }{\left(\frac{\partial }{\partial \theta }\log f\left(x;\theta \right)\right)}^{2}f\left(x;\theta \right)\;dx.$$

In our article, we consider the one-dimensional case by referring to the special (but often used in all kinds of applications) translation families, which satisfy the condition
f(x;θ)=f(x−θ).
In this case, using the identity
$$-\frac{\partial f}{\partial \theta }=\frac{\partial f}{\partial x},$$
we can write the definitions of Fisher information as follows:$$I_{F}=\int_{-\infty }^{\infty }f\left(x\right){\left(\frac{d}{dx}\ln f\left(x\right)\right)}^{2}dx\;.$$
By substitution $f\left(x\right):=\psi^{2}\left(x\right)$, we obtain
(1)$$I_{F}=4\int_{-\infty }^{\infty }{\left(\frac{d}{dx}\psi \left(x\right)\right)}^{2}dx\;.$$

Suppose we are looking for a real wave function ψ(x) that minimizes the value of the Fisher information under the following conditions:$$1\;=\;\int_{-\infty }^{\infty }\psi^{2}\left(x\right)dx,\;\;\;m=\int_{-\infty }^{\infty }x\;\psi^{2}\left(x\right)dx,\;\;\;r=\int_{-\infty }^{\infty }{\left(x-m\right)}^{2}\;\psi^{2}\left(x\right)dx\;.$$
It comes down to finding the minimum of the functional:$$\int_{-\infty }^{\infty }F\left(\psi \left(x\right),\frac{d}{dx}\psi \left(x\right),x\right)\;dx=\int_{-\infty }^{\infty }4{\left(\frac{d}{dx}\psi \left(x\right)\right)}^{2}dx-\int_{-\infty }^{\infty }\left(a\;+\;b\;x\;+\;c\;x^{2}\right)\psi^{2}\left(x\right)dx,$$
where *a*, *b*, *c* are Lagrange multipliers. The solution ψ(x) to the above variational problem defines the probability distribution $\psi^{2}\left(x\right)$ of a random variable *x* with a certain mean *m* and risk *r* for the minimum value of Fisher information.

It turns out that the above variational problem leads to solutions to the equation
(2)$$-\frac{1}{2\omega }\frac{\partial^{2}\psi }{\partial x^{2}}+\frac{\omega }{2}{\left(x-m\right)}^{2}\;\psi \;=\;\varepsilon \;\psi ,$$
where ω, ε, and *m* are constants used to parametrize Lagrange multipliers ($a=8\;\varepsilon \;\mu -4\;x_{0}^{2}\;\mu^{2}$, $b=8\;x_{0}\;\mu^{2}$, $c=-4\;\mu^{2}$ and translation $x\mapsto x+x_{0}-m$). For the detailed derivation of the above equation, readers are referred to [10]. This is the popular Schrödinger-type equation for a quantum harmonic oscillator. This is one of the basic equations of non-relativistic quantum mechanics. It plays a key role in many fields such as quantum optics and solid-state physics, and it is also the basis of modern chemistry. It is worth noting that this equation is derived with minimal assumptions. Only the definition of the Fisher information and assumptions about the real value of ψ(x) were used.

The solutions of Equation (Equation 2) form a discrete set of functions:(3)$$\psi_{n}\left(x\right)=\sqrt{\frac{\sqrt{\omega }}{2^{n}n!\sqrt{\pi }}}\;e^{-\frac{\omega \;{\left(x-m\right)}^{2}}{2}}\;H_{n}\left(\sqrt{\omega }\left(x-m\right)\right),$$
where $H_{n}\left(x\right)$ is the *n*–th Hermite polynomial. This complete orthonormal set of functions stretches the vector space $L^{2}$ over the field C square-integrable functions [14]. We identify the $\psi_{n}\left(x\right)$ functions as market strategies that determine the supply and demand curves [6]. The eigenvalues corresponding to the above eigenvectors take the form
$$\varepsilon =\varepsilon_{n}=n+\frac{1}{2},$$
for n=0,1,2, … The above quantity represents the total (supply and demand) risk observed as a result of the measurement of strategy $\psi_{n}\left(x\right)$. Moreover,
$$I_{F_{n}}=4\mu \varepsilon_{n},$$
where $I_{F_{n}}$ denotes the Fisher information carried by the strategy $\psi_{n}\left(x\right)$ [13].

Market strategies with minimal Fisher information may better reflect the specifics of the market. There are at least two sorts of arguments for minimizing information about markets.

If we have no information on the measures of probabilities of elementary events then we should treat all of them on the same footing (i.e., as equivalent). This argument is based on the famous Laplace principle of indifference.One should not expect anything else—more information involves higher costs on the revealing information side (*information is physical* [15]). This argument is based on the *no free lunch* principle.

It turns out there is an important relationship between Fisher information and risk [13]. The risk associated with a given strategy is the sum of the risk in its demand and supply representation. Note that the left side of the Equation (Equation 2) is the sum of the supply operator $R_{d}:=x^{2}$ and $-\frac{\partial^{2}}{\partial x^{2}}$. The operator $-\frac{\partial^{2}}{\partial x^{2}}$ is in fact the supply risk operator $R_{s}$. It results directly from the properties of the Fourier transform
$$\begin{array}{cc} -\frac{d^{2}\psi \left(x\right)}{dx^{2}}\overset{FT}{\to }y^{2}\;\psi \left(y\right) & ,\;\;\;\;\;\;\;\;\;\;\;\;\;\;\;\;x^{2}\;\psi \left(x\right)\overset{FT}{\to }-\frac{d^{2}\psi \left(y\right)}{dy^{2}}\;, \end{array}$$
where ψ(y):=FTψ(x). Therefore, the Fourier transform of Equation (Equation 2) leads to an equation of the same type. We have a connection between risk and information associated with strategy.

Hence, the Equation (Equation 2) is called risk balance equation [10] and lies at the root of the so-called the transactional interpretation for the principle of minimum Fisher information [13]. The left side of the Equation (Equation 2) is the the total risk operator (observable of market risk) of the buy–sell cycle.

## 3. The State Minimizing the Position-Momentum Uncertainty Principle

A natural continuation of analyses of a quantum description of the market in the context of minimum Fisher information is to examine the validity of describing market strategies using coherent states, in particular, squeezed coherent states.

In quantum mechanics, coherent states are the quantum states of a harmonic oscillator whose dynamics most closely resemble the oscillatory behaviour of a classical harmonic oscillator. Therefore, coherent states may better reflect the description of real financial markets, where we do not observe quantum effects. Furthermore, they are minimum uncertainty states for which the product of standard deviations of position and momentum measurements has the smallest value (they minimize Heisenberg’s uncertainty principle). In the quantum description of the market, we equate these two quantities with supply and demand. In such a supply-and-demand description of the market, the strategy that minimizes the uncertainty principle of position and momentum (in the market interpretation, supply and demand), i.e., the coherent state, can be treated as the most predictable when measuring market transactions.

As we mentioned before, we take Fisher information as a measure of the information carried by market strategies. It plays a fundamental role in the theory of estimation, which is reflected in the Cramér–Rao bound [16]
$$I_{F_{\theta }}\cdot \mathrm{var}\left(\hat{\theta }\right)⩾1\;.$$
The above inequality expresses a lower bound on the variance of unbiased estimators $\hat{\theta }$ of parameter θ. The symbol $I_{F_{\theta }}$ means Fisher information.

We come across a similar type of inequality in physics. These are the famous uncertainty principles mentioned above. The principles say that there are pairs of quantities that cannot be measured with any precision at the same time. Their mathematical form is as follows (see also [17,18]):(4)$$\Delta_{M}A\cdot \Delta_{M}B\geq \frac{\hbar }{2}\left|{\langle {\left\{A,B\right\}}_{\hbar }\rangle }_{M}\right|\;.$$
where

*A*, *B*—any observable,$\Delta_{M}A$, $\Delta_{M}B$—the uncertainties of, respectively, *A* and *B* observables in the *M* state,${\left\{A,B\right\}}_{\hbar }:=\frac{1}{i\hbar }\left(AB-BA\right)$—the quantum Poisson bracket and *ℏ*—the Planck constant.

The left side of the inequality (Equation 4) is minimized by coherent states. These are the basic objects we deal with in this article; therefore, we will derive the above inequality.

Let *A* and *B* denote any observables. By the inner product properties, we can write:$$\begin{array}{cc} \langle (A+i\alpha B)\psi |(A+i\alpha B)\psi \rangle & =\langle A^{2}\psi |\psi \rangle +\alpha^{2}\langle B^{2}\psi |\psi \rangle +i\alpha \langle \psi (AB-BA)|\psi \rangle \\ \; & =\langle A^{2}\psi |\psi \rangle +\alpha^{2}\langle B^{2}\psi |\psi \rangle -\alpha \hbar \langle \psi {\left\{A,B\right\}}_{\hbar }|\psi \rangle . \end{array}$$
Since for any α∈R
(5)$$0\leq \langle (A+i\alpha B)\psi |(A+i\alpha B)\psi \rangle ,$$
we have
(6)$$0\leq \langle A^{2}\psi |\psi \rangle +\alpha^{2}\langle B^{2}\psi |\psi \rangle -\alpha \hbar \langle \psi {\left\{A,B\right\}}_{\hbar }|\psi \rangle \;.$$

Let $P_{v}:=|v\rangle \langle v|$ and $M:=\sum_{k}p_{k}P_{\psi_{k}}$. By adding (with the weights $p_{k}$) the inequalities of the form (Equation 6), we have
(7)$$0\leq {\langle A^{2}\rangle }_{M}+\alpha^{2}{\langle B^{2}\rangle }_{M}-\alpha \hbar {\langle {\left\{A,B\right\}}_{\hbar }\rangle }_{M},$$
where ${\langle A\rangle }_{M}:=TrAM$.

In order for the above inequality to be satisfied for every α, it is necessary and sufficient that the discriminant of the second degree polynomial is non-positive, i.e.,
$$\Delta =\hbar^{2}{\langle {\left\{A,B\right\}}_{\hbar }\rangle }_{M}^{2}-4{\langle A^{2}\rangle }_{M}{\langle B^{2}\rangle }_{M}\leq 0\;.$$
Hence
(8)$$4{\langle A^{2}\rangle }_{M}{\langle B^{2}\rangle }_{M}\geq \hbar^{2}{\langle {\left\{A,B\right\}}_{\hbar }\rangle }_{M}^{2}\;.$$
Substituting $A-{\langle A\rangle }_{M}⟶A$ and $B-{\langle B\rangle }_{M}⟶B$ into (Equation 8), we obtain
(9)$$\Delta_{M}A\cdot \Delta_{M}B\geq \frac{\hbar }{2}\left|{\langle {\left\{A,B\right\}}_{\hbar }\rangle }_{M}\right|,$$
where $\Delta_{M}A:=\sqrt{Tr{\left(A-{\langle A\rangle }_{M}\right)}^{2}M}$. The minimum of the left side of the inequality (Equation 5) is achieved when
(10)$$\left(A+i\alpha B\right)|\psi \rangle =0\;.$$
Let us define the position and momentum operators
$$A|\psi (x)\rangle =\left(x-\mu \right)|\psi (x)\rangle ,\;\;\;\;\;\;B|\psi \rangle =-\frac{i}{\alpha }\sigma^{2}\frac{d\psi }{dx}\;.$$
The wave function solving the Equation (Equation 10) is well defined due to the phase factor $e^{i\nu x}$. The general form of the $\psi_{\mu ,\nu ,\sigma }$ state that minimizes the left side of the position-momentum uncertainty principle is of the form
(11)$$\psi_{\mu ,\nu ,\sigma }=\frac{1}{\sqrt{\sqrt{\pi }\sigma }}e^{i\nu x}e^{-\frac{{\left(x-\mu \right)}^{2}}{2\sigma^{2}}}.$$
This is the so-called squeezed coherent state formed from the deformation of the coherent state involving the broadening (or narrowing) of its wave function. The Function (Equation 11) is a Gaussian function. The Fourier transform of this type of function is also a Gaussian function but with a different width, i.e., if $\sigma^{2}$ is large/small then the graph of the Fourier transform is narrow/broad. This property is reflected in signal analysis. The more concentrated a signal is in the time domain, the more spread out it is in the frequency domain. To be more precise, the product of measure of signal duration and the corresponding measure of the width of its frequency spectrum is bounded from below. This property of the Fourier transform is closely related to the uncertainty principle in quantum mechanics. For us, it will be crucial in describing the total risk of the squeezed coherent strategy, which we will address later in the paper.

At the end of this paragraph, it is worth noting that coherent states are present in many research topics covering such fields of physics as quantum mechanics, optics, quantum chemistry, atomic physics, statistical physics, nuclear physics, particle physics, and cosmology. Recently, they have also found practical application in quantum communication, the distribution of quantum keys, and quantum digital signatures [19]. All this confirms the importance of coherent states for the development of various scientific theories.

## 4. Squeezed Coherent State and the Eigenstates of Quantum Harmonic Oscillator

Without losing the generality of our considerations and in order to obtain greater clarity of results, we can assume ω=1 and m=0. From now on, by the eigenstate of the Hamiltonian of the quantum oscillator (Equation 3) we will understand the state of the form
(12)$$\psi_{n}\left(x\right)=\frac{1}{\sqrt{2^{n}n!\sqrt{\pi }}}H_{n}\left(x\right)\;e^{-\frac{x^{2}}{2}}.$$
The set of all vectors (Equation 12) forms the orthogonal basis of the space $L^{2}$ of square-integrable functions over the field C. We can represent (Equation 11) on this basis. For this, we will find the inner product $\langle \psi_{n}|\psi_{\mu ,\nu ,\sigma }\rangle$.

The application of the exponential generating function of Hermite polynomials
$$g\left(x,t\right)=e^{-t^{2}+2tx}$$
provides
$$\langle \psi_{n}|\psi_{\mu ,\nu ,\sigma }\rangle =\frac{1}{\sqrt{\pi 2^{n}n!\sigma }}\frac{d^{n}}{dt^{n}}{|}_{t=0}e^{-t^{2}}\int_{-\infty }^{\infty }e^{(2t+i\nu )x}e^{-\frac{{\left(x-\mu \right)}^{2}+\sigma^{2}x^{2}}{2\sigma^{2}}}dx\;.$$
The function under the integral contains the quadratic polynomial of the variable *x* in the exponent
$$-\left(\sigma^{2}+1\right)x^{2}+\left(2\sigma^{2}\left(2t+i\nu \right)+2\mu \right)x-\mu^{2}$$
which we can rewrite as follows:$$-\left(\sigma^{2}+1\right)\left(x+\frac{2\sigma^{2}\left(2t+i\nu \right)+2\mu }{-2(\sigma^{2}+1)}\right)+\frac{{\left(\sigma^{2}\left(2t+i\nu \right)+\mu \right)}^{2}}{2\sigma^{2}+1}-\mu^{2}\;.$$
Since for u>0
$$\int_{-\infty }^{\infty }e^{-ux^{2}}dx=\sqrt{\frac{\pi }{u}},$$
it follows that
(13)$$\begin{array}{cc} \langle \psi_{n}|\psi_{\mu ,\nu ,\sigma }\rangle & =\frac{1}{\sqrt{2^{n-1}n!\frac{\sigma^{2}+1}{\sigma }}}\frac{d^{n}}{dt^{n}}{|}_{t=0}e^{-t^{2}}e^{\frac{4\sigma^{2}t^{2}+4i\nu \sigma^{2}t+4\mu t-\nu^{2}\sigma^{2}+2i\mu \nu -\mu^{2}}{2(\sigma^{2}+1)}}\\ \; & =\frac{1}{\sqrt{2^{n-1}n!\frac{\sigma^{2}+1}{\sigma }}}e^{\frac{-\nu^{2}\sigma^{2}+2i\mu \nu -\mu^{2}}{2(\sigma^{2}+1)}}\frac{d^{n}}{dt^{n}}{|}_{t=0}e^{\frac{\sigma^{2}-1}{\sigma^{2}+1}t^{2}+2t\frac{i\nu \sigma^{2}+\mu }{\sigma^{2}+1}}. \end{array}$$
The final form of Formula (Equation 13) depends on making an additional assumption about the value of σ parameter.

We consider three cases as follows:If σ=1 there would be
$$\langle \psi_{n}|\psi_{\mu ,\nu ,\sigma =1}\rangle =\frac{1}{\sqrt{n!}}e^{\frac{-\nu^{2}+2i\mu \nu -\mu^{2}}{4}}{\left(\frac{\mu +i\nu }{\sqrt{n}}\right)}^{n}$$If σ<1 there would be
$$\begin{array}{c} \langle \psi_{n}|\psi_{\mu ,\nu ,\sigma <1}\rangle =\frac{1}{\sqrt{n!}}\sqrt{\frac{2\sigma }{1+\sigma^{2}}}e^{\frac{-\nu^{2}\sigma^{2}+2i\mu \nu -\mu^{2}}{2(\sigma^{2}+1)}}{\left(\frac{1-\sigma^{2}}{2(1+\sigma^{2})}\right)}^{\frac{n}{2}}H_{n}\left(\frac{\mu +i\nu \sigma^{2}}{\sqrt{\left(1-\sigma^{2}\right)\left(1+\sigma^{2}\right)}}\right) \end{array}$$
It follows from the following formula
$${\left.\frac{d^{n}}{dt^{n}}\right|}_{t=0}e^{\frac{\sigma^{2}-1}{\sigma^{2}+1}t^{2}+2t\frac{i\nu \sigma^{2}+\mu }{\sigma^{2}+1}}={\left.\left(\frac{1-\sigma^{2}}{1+\sigma^{2}}\right)\frac{d^{n}}{d\tau^{n}}\right|}_{\tau =0}e^{-\tau^{2}+2\tau \frac{i\nu \sigma^{2}+\mu }{\sqrt{\left(1-\sigma^{2}\right)\left(1+\sigma^{2}\right)}}},\;$$
where $\tau :=\sqrt{\frac{1-\sigma^{2}}{1+\sigma^{2}}}t$.If σ>1 there would be
$$\langle \psi_{n}|\psi_{\mu ,\nu ,\sigma >1}\rangle =\frac{1}{\sqrt{n!}}\sqrt{\frac{2\sigma }{\sigma^{2}}+1}e^{\frac{-\nu^{2}\sigma^{2}+2i\mu \nu -\mu^{2}}{2(\sigma^{2}+1)}}{\left(\frac{\sigma^{2}-1}{2(\sigma^{2}+1)}\right)}^{\frac{n}{2}}i^{n}H_{n}\left(\frac{\nu \sigma^{2}-i\mu }{\sqrt{\left(\sigma^{2}-1\right)\left(\sigma^{2}+1\right)}}\right)\;.$$
It follows from the following formula
$${\left.\frac{d^{n}}{dt^{n}}\right|}_{t=0}e^{\frac{\sigma^{2}-1}{\sigma^{2}+1}t^{2}+2t\frac{i\nu \sigma^{2}+\mu }{\sigma^{2}+1}}=i^{n}{\left.\left(\frac{\sigma^{2}-1}{\sigma^{2}+1}\right)\frac{d^{n}}{d\tau^{n}}\right|}_{\tau =0}e^{-\tau^{2}+2\tau \frac{i\nu \sigma^{2}-i\mu }{\sqrt{\left(\sigma^{2}-1\right)\left(\sigma^{2}+1\right)}}},$$
where $\tau :=i\sqrt{\frac{\sigma^{2}-1}{\sigma^{2}+1}}\;t.$

## 5. The Generalized Poisson Distribution

Let us determine the probability $p_{n}:={\left|\langle \psi_{\mu ,\nu ,\sigma }|\psi_{n}\rangle \right|}^{2}$ that the state $\psi_{\mu ,\nu ,\sigma }$ will be found in the state $\psi_{n}$.

If we take σ=1, we obtain
$$p_{n}=\frac{1}{n!}e^{-\frac{\mu^{2}+\nu^{2}}{2}}{\left(\frac{\mu^{2}+\nu^{2}}{2}\right)}^{n}\;.$$
This is the standard Poisson distribution.If σ<1, then
$$p_{n}=\frac{1}{n!}\frac{2\sigma }{1+\sigma^{2}}e^{-\frac{\nu^{2}\sigma^{2}+\mu^{2}}{\sigma^{2}+1}}{\left(\frac{1-\sigma^{2}}{2(1+\sigma^{2})}\right)}^{n}H_{n}\left(\frac{\mu +i\nu \sigma^{2}}{\sqrt{1-\sigma^{4}}}\right)H_{n}\left(\frac{\mu -i\nu \sigma^{2}}{\sqrt{1-\sigma^{4}}}\right)\;.$$
By Mehler’s formula, we obtain the moment-generating function:
$$\sum_{n=0}^{\infty }p_{n}e^{sn}=\frac{2\sigma }{1+\sigma^{2}}\;e^{-\frac{\nu^{2}\sigma^{2}+\mu^{2}}{\sigma^{2}+1}}\frac{1}{\sqrt{1-t^{2}}}e^{2\frac{vt-wt^{2}}{1-t^{2}}}{|}_{v=\frac{\mu^{2}+\nu^{2}\sigma^{4}}{1-\sigma^{4}},w=\frac{\mu^{2}-\nu^{2}\sigma^{4}}{1-\sigma^{4}},t=\frac{1-\sigma^{2}}{1+\sigma^{2}}e^{s}}\;.$$If σ>1, then
$$p_{n}=\frac{1}{n!}\frac{2\sigma }{\sigma^{2}+1}e^{-\frac{\nu^{2}\sigma^{2}+\mu^{2}}{\sigma^{2}+1}}{\left(\frac{\sigma^{2}-1}{2(\sigma^{2}+1)}\right)}^{n}H_{n}\left(\frac{\nu \sigma^{2}+i\mu }{\sqrt{\sigma^{4}-1}}\right)H_{n}\left(\frac{\nu \sigma^{2}-i\mu }{\sqrt{\sigma^{4}-1}}\right)\;.$$
The moment-generating function corresponding to the above probability distribution is of the form
$$\sum_{n=0}^{\infty }p_{n}e^{sn}=\frac{2\sigma }{\sigma^{2}+1}e^{-\frac{\nu^{2}\sigma^{2}+\mu^{2}}{\sigma^{2}+1}}\frac{1}{\sqrt{1-t^{2}}}e^{2\frac{vt-wt^{2}}{1-t^{2}}}{|}_{v=\frac{\mu^{2}+\nu^{2}\sigma^{4}}{\sigma^{4}-1},w=\frac{\nu^{2}\sigma^{4}-\mu^{2}}{\sigma^{4}-1},t=\frac{\sigma^{2}-1}{\sigma^{2}+1}e^{s}}\;.$$

It is worth noting that in case 2 and 3, the moment-generating functions are the same. Hence, the conclusions presented in the next paragraph apply to both of these cases. Let us call the obtained distribution as the generalized Poisson distribution.

## 6. Risk and the Risk of Risk

Let us recall that the starting point of our analysis was to take Fisher information as a measure of the information carried by market strategies $\psi \in L^{2}$. The Fisher information has a wide range of applications, from the optimal design of experiments and its relationship to the laws of physics [11,12,20], to the role it plays in estimation theory and statistics [16]. We adopted the variance of a random variable (in our case, the logarithm of the price) with a probability density $\psi^{2}$ as a measure of the risk associated with the market strategy. Next, we posed the question: which probability distributions determined by market strategies ψ produce, at a given variance, the minimum Fisher information? The search for an answer to this question leads to a one-dimensional quantum harmonic oscillator equation. The solutions to this equation determine our market strategies with the minimum Fisher information. They form an orthogonal complete basis of the space $L^{2}$, which allows us to write any strategy from $L^{2}$ in this basis. Its elements are the eigenvectors of the total risk operator—the left-hand side of the equation (Equation 2). These strategies have the best defined risk as the variance of the risk operator in these states is 0.

Let us denote by *R* the the total risk operator. By the total risk associated with a given strategy ψ, we understand the expectation value of *R* in the state ϕ, denoted as ${\langle R\rangle }_{\phi }$. The *R* operator has a complete set of eigenvectors $\psi_{n}$, with eigenvalues $\varepsilon_{n}=n+\frac{1}{2}$, for n=0,1,2,…. The expectation value of *R* can be expressed as
(14)$${\langle R\rangle }_{\phi }=\sum_{n=0}^{\infty }\varepsilon_{n}{\left|\langle \phi |\psi_{n}\rangle \right|}^{2}\;.$$
The above formula determines the total risk carried by the market strategy ϕ.

In this article, we focus on the $\psi_{\mu ,\nu ,\sigma }$ strategies, which are provided by the Formula (Equation 11). They are the so-called squeezed coherent states. These are specific strategies because of the position-momentum uncertainty principle minimizing property. In the previous paragraph, we determined the probability of finding these strategies in the eigenstates of the total risk operator.

The resulting generalized Poisson distribution determines the relationship between the family of all squeezed coherent states and their image in terms of quantum risk. Its expectation value
(15)$${\langle R\rangle }_{\psi_{\mu ,\nu ,\sigma }}=\frac{1}{2}\mu^{2}+\frac{1}{4}\sigma^{2}+\frac{1}{2}\nu^{2}+\frac{1}{4}\sigma^{-2},$$
determines the total risk of the strategy (Equation 11). The above formula can be obtained by the property of the moment-generating function of distribution of the random variable *n* determined in Section 5. Here, we consider the $n+\frac{1}{2}$ random variable, and its expected value is greater than the expected value of the *n* random variable by $\frac{1}{2}$. The higher-order moments are the same for both of these random variables.

By the definition of the Fourier transform, we have
$$\hat{f}\left(y\right):=\int_{-\infty }^{\infty }f\left(x\right)\;e^{-iyx}dx=\sqrt{2\pi }\sigma e^{-i\mu y}e^{\frac{-\sigma^{2}{\left(y-\nu \right)}^{2}}{2}},$$
where $\hat{f}$ is the Fourier transform of
$$f\left(x\right)=e^{i\nu x}e^{\frac{-{\left(x-\mu \right)}^{2}}{2\sigma^{2}}}\;.$$

Hence, the demand and supply representations of the strategy (Equation 11) are of the same form (but not equal). The role of the μ and ν parameters has changed, i.e., μ→ν and ν→μ and $\sigma \to \sigma^{-1}$. Therefore, the expression (Equation 15) represents both the supply and demand parts of the total risk of the market strategy.

The μ, ν, and σ parameters define the strategies (Equation 11) and the total risk associated with this strategy, expressed by the Formula (Equation 15). The uncertainty of this risk (let us call it the risk of risk) is the second central moment of the $\varepsilon_{n}$ random variable with the distribution determined by our sqeezed coherent state. The risk of risk formula in our case is as follows:(16)$$\frac{\mu^{2}\sigma^{2}+\nu^{2}\sigma^{-2}}{2}+\frac{\sigma^{4}+\sigma^{-4}-2}{8}\;.$$
The above expression takes the smallest value for μ=ν=0 and σ=1, that is, when the wave function of the strategy is the oscillator ground state.

Strategies (Equation 11) for which the estimation of the moments of the distribution carries the minimum information about their parameters μ,ν,σ seem to be particularly important. The average profit of a strategy (see profit intensity [21]) is determined by the parameters μ and ν. If we assume one of them (e.g., μ) is equal to 0 and select a small enough $\sigma^{-1}$ (if ν=0 then small should be σ), then the moment (Equation 16) will not carry information about the parameter ν (respectively μ). Such squeezed coherent states are characterized by the flattened error function of demand (buying equally willingly at any logarithm of price) and the Heaviside step function of supply (or vice versa). Determining the value of the parameter ν (or μ) that maximizes the profit intensity obtained by such strategies is described in the paper [21] and involves a unique extreme property of the profit intensity function at its fixed point [22].

## 7. Conclusions

The construction of a complete theory of the quantum market should respect all the limitations of quantum mechanics. Examples of such constraints are those established by the famous principles of uncertainty (hence the need to study such strategies as coherent strategies that minimize the uncertainty principle).

In the article, we analyzed the risks carried by this type of strategy. We introduced the squeezed coherent strategies’ risk of risk concept in their transactional interpretation. This is their essential numerical characteristic. After all, the risk of a squeezed-coherent strategy modelled on a physical object realizing the physical system of a quantum oscillator is the energy of that oscillator state. Therefore, the risk of risk of such a strategy is the square of the uncertainty of its energy, and through the famous uncertainty relation for time and energy [23] sets a lower limit for the uncertainty of time, usually interpreted as the lifetime of the perturbation that is this impermanent oscillator state, or the preparation time of this state. The necessary lifetime of a strategy operating in the market is inversely proportional to the intensity of profits achievable by it. The shorter the transactions last, the higher the profits.

## Data Availability

Not applicable.

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
