# Peer review of "Transactional Interpretation and the Generalized Poisson Distribution"

_entropy, 2022, doi:10.3390/e24101416_

Round 1

Reviewer 1 Report

  •  
  • Review in the attached file.

Reviewer 2 Report

Review of: “Transactional Interpretation and the generalized Poisson distribution” Makowski & Piotrowski 

In this manuscript, the authors applied the minimum Fisher information principle to the study of the quantum-like approach to the description of the market. Specifically, 

a squeezed state was employed for the market strategy, and its probability with respect to the eigenstates of market risk was derived, proven to be of a generalized Poisson distribution. In fact, this manuscript is a follow-up of their long-line research on quantum-like financial market.

Let me ask questions, though:  

1. English should be improved, e.g., Page 1, Abstract, “... is to continue the study on the quantum-like approach the description of the market ...” --> I would suggest: “... is to study the quantum-like approach to the description of the market ...”

“... we focus on representation of any squeezed coherent ...” --> “... we focus on the representation of any squeezed coherent ...”

Page 3, the 2nd last paragraph of Section 2, “Therefore, the Fourier transform of Equation (2) lead to an ...” --> “Therefore, the Fourier transform of Equation (2) leads to an ...”

2. Page 2, after Eq. (2), “... where \omega i \epsilon ...”? 

3. Page 3, after Eq. (3), “We identify the \psi_n(x) functions as market strategies that determine the supply and demand curves.” YES. However, I'm still wondering about a financial market meaning of the associated quantity \epsilon_n itself. How can this be interpreted, especially \epsilon_0 (from lack of my knowledge)?

4. Sections 4-6 discuss their new findings of this manuscript ... 

Page 8, the 3rd last paragraph of Section 6, “The role of \mu i \nu parameters is changed, i.e., \mu \to \nu i \nu \to \mu and ...”?

Let me make the final decision after receiving the authors’ reply to the afore-stated questions.

Round 2

Reviewer 1 Report

The paper benefited from the improvements made by the authors. I also accept their explanation concerning more detailed discussion of harmonic oscillator etc. However, following the same line of thought, they should add more information concerning the main ideas of „quantum” market. Then the paper will become acceptable for publication.

Reviewer 2 Report

This manuscript is now qualified for publication in Entropy.
